# Separation of Hydrochloric Acid and Oxalic Acid from Rare Earth Oxalic Acid Precipitation Mother Liquor by Electrodialysis

**DOI:** 10.3390/membranes13020162

**Published:** 2023-01-27

**Authors:** Hengcheng Zhou, Peihai Ju, Shaowei Hu, Lili Shi, Wenjing Yuan, Dongdong Chen, Yujie Wang, Shaoyuan Shi

**Affiliations:** 1College of Resources and Environment, Nanchang University, Nanchang 330031, China; 2Ganjiang Innovation Academy, Chinese Academy of Sciences, Ganzhou 341119, China; 3Jiangxi Province Key Laboratory of Cleaner Production of Rare Earths, Ganzhou 341119, China; 4Technology Center of Angang Steel Co., Ltd., Anshan 114009, China; 5Institute of Process Engineering, Chinese Academy of Sciences, Beijing 100190, China

**Keywords:** acid separation, precipitation mother, electrodialysis, selective anion exchange membrane, membrane fouling

## Abstract

In this study, the hydrochloric acid from rare earth oxalic acid precipitation mother liquor was separated by electrodialysis (ED) with different anion exchange membranes, including selective anion exchange membrane (SAEM), polymer alloy anion exchange membrane (PAAEM), and homogenous anion exchange membrane (HAEM). In addition to actual wastewater, nine types of simulated solutions with different concentrations of hydrochloric acid and oxalic acid were used in the experiments. The results indicated that the hydrochloric acid could be separated effectively by electrodialysis with SAEM from simulated and real rare earth oxalic acid precipitation mother liquor under the operating voltage 15 V and ampere 2.2 A, in which the hydrochloric acid obtained in the concentrate chamber of ED is of higher purity (>91.5%) generally. It was found that the separation effect of the two acids was related to the concentrations and molar ratios of hydrochloric acid and oxalic acid contained in their mixtures. The SEM images and ESD–mapping analyses indicated that membrane fouling appeared on the surface of ACS and CSE at the diluted side of the ED membrane stack when electrodialysis was used to treat the real rare earth oxalic acid precipitation mother liquor. Fe, Yb, Al, and Dy were found in the CSE membrane section, and organic compounds containing carbon and sulfur were attached to the surface of the ACS. The results also indicated that the real rare earth precipitation mother liquor needed to be pretreated before the separation of hydrochloric acid and oxalic acid by electrodialysis.

## 1. Introduction

In the rare earth industrial production process, it is inevitable to use a large amount of hydrochloric acid and oxalic acid for the dissolution and precipitation of rare earth [1]. During hydrometallurgical processing, REEs (Rare earth elements) are transferred from the solid phase into a solution for further concentration and purification [2,3]. After adequate enrichment and separation, high–purity rare earth salts are precipitated from a concentrated REE solution in the final processing stages. In the recovery step, because of the outstanding low solubility products of rare earth oxalates, large amounts of oxalic acid have been used to achieve satisfactory precipitation performance. Now oxalic acid is a commonly used precipitant in the purification of rare earth (RE) from a concentrated pregnant leach solution (PLS) [4], in which the H^+^–ions contained in oxalic acid are released and combined with Cl^−^ in a concentrated solution of RE to form hydrochloric acid with high concentration. Therefore, the rare earth oxalic acid precipitation mother liquor contains excessive oxalic acid, hydrochloric acid, low concentrations of residual rare earth elements, and other impurities. The effluents can only be discharged into natural water bodies after proper treatment; otherwise, the high COD and high acids contained in the waste liquid will cause serious environmental pollution.

At present, there are a few reports on the disposal and recovery of oxalic acid and hydrochloric acid from the rare earth oxalic acid precipitation mother liquor, including limestone neutralization [5], membrane distillation [6], evaporation crystallization [7]**,** and extraction [8]. However, these methods still have some disadvantages. For example, lime neutralization is a common method to treat acid wastewater at present [9]. However, this approach does not recover the resources of hydrochloric acid and oxalic acid and simultaneously generates a large amount of gypsum residue, which also easily causes blockage of the subsequent treatment system [10]. The performance of membrane distillation is limited by low temperature and concentration polarization leading to vapor pressure reduction [11]. Moreover, long-term operation of the membrane distillation system will lead to membrane scaling, pore clogging, and rate reduction due to the impurities contained in wastewater [12]. At present, due to the complex physical and chemical properties of the mother liquor containing high acids and other impurities, the separation and recovery of acids from the effluents is still not ideal using existing methods. It is critical to explore a new acid separation and recovery method for the rare earth oxalic acid precipitation mother liquor.

Electrodialysis technology is widely used in the desalination of brackish water, seawater, and industrial wastewater [13,14,15]. Another unique advantage of electrodialysis is the selective separation of the univalent ion from the multivalent ion (e.g., Cl^−^ versus SO_4_^2−^) with monovalent permselective AEMs due to their selective permeability for the two ions [16,17,18,19,20,21,22]. Furthermore, as promising sustainable processes [23], membrane processes including diffusion dialysis [24] and Nanofiltration membranes [25] were also considered in the recovery options of REEs from waste streams. Based on the research results reported above, in order to realize the resource treatment of rare earth oxalic acid precipitation mother liquor via acid recovery, the feasibility of separating oxalic acid and hydrochloric acid by electrodialysis was investigated first in this study. In order to evaluate the effect of membrane material on the separation performance, the separation performance of three different types of AEMs including SAEM, HAEM, and PAAEM for oxalic acid and hydrochloric acid was compared. The separation effect of hydrochloric acid and oxalic acid was also examined for the different acid concentrations and ratios. The hydrochloric acid contained in the real mother liquor from rare earth oxalic acid precipitation was separated using electrodialysis, and its membrane fouling was also analyzed and discussed. This study aims to explore a green and effective method for the separation and recovery of hydrochloric acid and oxalic acid via electrodialysis to achieve the resource treatment of mother liquor of rare earth oxalic acid precipitation.

## 2. Experiment

### 2.1. Materials

The three electrodialysis systems were composed of the ACS anion exchange membrane (SAEM) and the CSE cation exchange membrane obtained from ASTOM company, Tokyo, Japan. And all the other membranes including the AAM anion exchange membrane (PAAEM) and the CAM cation exchange membrane, the ATG–10 anion exchange membrane (HAEM), and the CTG–10 cation exchange membrane were obtained from the Lanran Company, Hanzhou, China.

The parameters of ion exchange membranes used in the experiments are shown in Table 1. The membranes were soaked in a 0.2 M sodium chloride solution (containing 0.4% (wt.) sodium bisulfite to inhibit the growth of bacteria and act as a reducing agent in this study) for 48 h before use and then cleaned with deionized water three times.

The reagents used were as follows: Sodium sulfate (Na_2_SO_4_ 99%), sodium bisulfite (NaHSO_4_ 99%), and Hydrochloric acid (HCl 36–38%) were obtained from Titan Scientific Co., Ltd, Shanghai, China. Oxalic acid dihydrate (H_2_C_2_O_4_·2H_2_O) was supplied by Sinopharm Chemical Reagent Co., Ltd, Shanghai, China. The standard chlorine solution came from the National Non–ferrous Metals and Electronic Materials Analysis and Testing Center, Beijing, China. The deionized water used in this study was obtained from the pure water machine of Kerefeng Company, Xiamen, China. The conductivity of the product water was 0.3 μs/cm.

### 2.2. Experimental Device and Operation

The electrodialysis device (EX–3BT) used in the experiments came from the Lanran Company, Hangzhou, China. Three circulating loops were formed by a peristaltic pump driving the different solutions, respectively, through the electrode, dilute, and concentrate chambers. The concentrate chamber was initially fed 500 mL of deionized (DI) water. Different concentrations of mixed hydrochloric acid and oxalic acid were fed into the dilute chamber as feed solutions. The electrode chamber was rinsed with 500 mL 4% (wt.) sodium sulfate. Each chamber solution was circulated at 1000 mL/min driven by a peristaltic pump. Before and after each experiment, the three chambers should be cleaned three times using DI water. The schematic diagram of the electrodialysis device is shown in Figure 1.

### 2.3. Analysis

Ion chromatography (Thermo Fisher Scientific, Milwaukee, WI, USA.) was used to detect the concentration of chloride ionic contained in the solutions of concentrate and dilute chambers. A TOC–2000 instrument (Metash Instruments Co., Ltd., Shanghai, China) was used to measure the concentration of oxalic acid in both chambers. The AEMs used before and after were characterized by Fourier Transform infrared spectroscopic analysis (ATR–FTIR, Thermo Fisher IS5, Plainville, MA, USA). Surface micro–morphology of the membranes was observed via scanning electron microscopy (SEM, TESCAN MIRA LMS, Brno, Czech Republic). To examine the distribution of the elements on the membranes, EDS was used to obtain the element mapping at an accelerating voltage of 15 kV with a magnification of 20 K. The roughness of different membranes was analyzed using atomic force microscopy (AFM, Bruker Dimension ICON, Billerica, MA, USA).

Ion migration was determined according to the relationship between the ion concentration and solution volume, and the specific evaluation indexes included the ion migration quantity, transport number, and selective transmission coefficient [26].

The amount of migration of the Cl^−^ ion and the oxalate ion (OA^−^) was calculated as follows:(1)$$\Delta n_{Cl^{-}/OA^{-}}=\frac{\left|\Delta n_{dc}{}_{\left(Cl^{-}/OA^{-}\right)}\right|+\left|\Delta n_{cc(Cl^{-}/OA^{-})}\right|}{2}$$
where $\Delta n_{Cl^{-}/OA^{-}}$ is the Cl^−^ or oxalate ion migration amount (*meq)*, $\Delta n_{dc(Cl^{-}/OA^{-})}$ is the ion migration amount of Cl^−^ or the oxalate ion in the dilute chamber, and $\Delta n_{cc(Cl^{-}/OA^{-})}$ is the ion migration amount of Cl^−^ or the oxalate ion in the concentrate chamber. They can be calculated using the following equation.
(2)$$\Delta n_{dc}{}_{\left(Cl^{-}/OA^{-}\right)}=C_{dc(Cl^{-}/OA^{-})}V_{dc}{}_{\left(Cl^{-}/OA^{-}\right)}-C_{0}{}_{\left(Cl^{-}/OA^{-}\right)}V_{0(Cl^{-}/OA^{-})}$$
(3)$$\Delta n_{cc}{}_{\left(Cl^{-}/OA^{-}\right)}=C_{cc(Cl^{-}/OA^{-})}V_{cc}{}_{\left(Cl^{-}/OA^{-}\right)}-C_{0}{}_{\left(Cl^{-}/OA^{-}\right)}V_{0(Cl^{-}/OA^{-})}$$
where $C_{dc(Cl^{-}/OA^{-})}$ and $C_{cc(Cl^{-}/OA^{-})}$ are the concentrations of Cl^−^ or the oxalate ion in the dilute chamber and the concentrate chamber at a certain time, respectively. $V_{dc(Cl^{-}/OA^{-})}$ and $V_{cc(Cl^{-}/OA^{-})}$ are the volumes of Cl^−^ or the oxalate ion solution in the dilute chamber and the concentrate chamber at a certain time, respectively. $C_{0(Cl^{-}/OA^{-})}$ and $V_{0(Cl^{-}/OA^{-})}$ are the ions of Cl^−^ or the oxalate ion concentration and volume of the dilute chamber at the initial time, respectively.

The transport number *t* represents the percentage of each ion in the total amount of migrated ions and was defined as follows:(4)$$t_{OA^{-}}=\frac{\Delta n_{OA^{-}}}{\Delta n_{OA^{-}}+\Delta n_{Cl^{-}}}$$
(5)$$t_{Cl^{-}}=\frac{\Delta n_{Cl^{-}}}{\Delta n_{OA^{-}}+\Delta n_{Cl^{-}}}$$

The ratio of *t_Cl_^−^
*to *t_OA_^−^* shows the Cl^−^ transport number to oxalate ion transport number (*t_Cl_^−^ /t_OA_^−^*), which can illustrate the selective permeability of AEM to Cl^−^. The larger the value is, the better the selective permeability of Cl^−^ is.

In order to better measure the hindering effect of ion exchange membrane on oxalic acid migration, the perm-selectivity of AEM between Cl^−^ to OA^−^ was calculated. It was defined as follows:(6)$$P_{Cl^{-}}^{OA^{-}}=\frac{t_{OA^{-}}/C_{0(OA^{-})}^{}}{t_{Cl^{-}}/C_{0(Cl^{-})}^{}}$$
where *t_OA^−^_* and *t_Cl_^−^
*represent the transport number of the oxalate ion and Cl^−^, respectively. $C_{0(OA^{-})}^{{}^{}}$ and $C_{0(Cl^{-})}^{{}^{}}$ represent the initial concentrations of the oxalate ion and Cl^−^ in the dilute chamber, respectively.

The smaller value of $P_{Cl^{-}}^{OA^{-}}$ indicated that fewer oxalate ions migrate across the AEM, which means the AEM has a better effect at hindering oxalate acid migration.

The hydrochloric acid purity (%*HCl*) obtained in the concentrate chamber is defined as its mol fraction of the two acids present as shown in Equation (7):(7)$$\%HCl=\frac{C_{HCl,t}^{}\times V_{t}^{}}{C_{HCl,t}^{}\times V_{t}^{}+C_{OA,t}^{}\times V_{t}^{}}\times 100\%$$
where *C_HCl,t_* and *C_OA,t_* refer to the final hydrochloric acid and oxalic acid concentration in the concentrate chamber after being treated, respectively. *V_t_* is the volume of the concentrate chamber after being treated.

## 3. Results and Discussion

### 3.1. Separation of HCl and OA by Different AEMs

The oxalic acid and hydrochloric acid were separated from their mixture solutions by ED with different ion exchange membranes, including SAEM, HAEM, and PAAEM. The concentrations of oxalic acid and hydrochloric acid in the two chambers were measured at the same time intervals, and the concentration changes of hydrochloric acid and oxalic acid in the different chambers are shown in Figure 2.

The results showed that SAEM had the best separation effect of OA and HCl among the three kinds of AEMs. In the ED experiment of the 2 M HCl + 0.4 M OA solution, the final oxalic acid concentrations in the concentrate chamber obtained was C_OA_ (SAEM, 54.63 mM) < C_OA_ (HAEM, 154.48 mM) < C_OA_ (PAAEM, 140.45 mM), while the final hydrochloric acid concentration in the concentrate chamber obtained was C_HCl_ (SAEM, 1288.56 mM) > C_HCl_ (HAEM, 1173.01 mM) > C_HCl_ (PAAEM, 1103.56 mM). The results indicated that, compared with hydrochloric acid, oxalic acid was easier to be maintained by SAEM, and the concentration of hydrochloric acid in the concentrate chamber was higher with the SAEM than that obtained with the other two AEMs. The concentration of oxalic acid in the dilute chamber at the end of electrodialysis was C_OA_ (SAEM, 338.01 mM) > C_OA_ (HAEM, 224.12 mM) > C_OA_ (PAAEM, 221.01 mM), which indicated that the concentration of oxalic acid remaining in the dilute chamber exhibited the opposite trend to that in the concentrate chamber. In the mixed acid solution of 3 M HCl + 0.4 M OA and 4 M HCl + 0.4 M OA, a similar change trend of acid concentrations in the different chambers was also obtained in the same electrodialysis units. This indicated that SAEM has the best performance for oxalic acid among the three AEMs. Therefore, the SAEM could be used for the separation of OA and HCl acids from their mixture via electrodialysis.

The SEM images shown in Figure 3a–c indicated that the surface micro–morphologies of the three AEMs are rather different. There were many small uniform particles and holes that appeared on the surface of HAEM, compared with the larger cracks on the surface of PAAEM. Due to the size sieving [27], the pore structure on the surface of PAAEM may be one of the reasons for its low oxalic acid interception. The pore structure on the PAAEM membrane may lead to more easy oxalic acid transport in the electrodialysis process, thus causing a low oxalic acid interception. Different from the other two AEMs, the surface of the SAEM is smoother and has no holes. The literature [28] reported a similar SAEM prepared by a “paste method” with a highly cross–linked layer deposited on its surface. This kind of cross–linked layer makes the SAEM surface smooth and uniform, which makes it suitable for the selective separation of ions with different valence states and radius sizes. Therefore, based on the differences in the valence states and radius sizes between Cl^−^ and the oxalate ion, electrodialysis with the SAEM achieved the efficient separation of hydrochloric acid and oxalic acid from their mixture, which could be attributed to the electrostatic repulsion and size-sieving effect [29,30].

According to the ATR-FTIR analyses in Figure 3d, compared with the other two kinds of AEMs, it was speculated that the surface of SAEM has a modified layer with more –P-O and –S=O functional groups. Both of these functional groups are negatively charged and have a larger electrostatic repulsion on oxalate than that on the chloride ion. It was assumed that the modified layer could result in better selective transmittance to Cl^−^ over oxalate ions, so the efficient separation of hydrochloric acid and oxalic acid in their mixture could be realized by the SAEM in the electrodialysis.

### 3.2. Effect of OA and HCl Ratio on SAEM Separation

#### 3.2.1. Concentration of Oxalic and Hydrochloric Acid in Each Chamber

The mixtures with different concentrations of hydrochloric acid and oxalic acid mixtures were separated by electrodialysis with the SAEM. The experimental data are shown in Figure 4.

Figure 4a,b shows the influence of different initial concentrations of oxalic acid in the dilute chamber on the separation effect of oxalic acid and hydrochloric acid when the concentration of hydrochloric acid in the dilute chamber was fixed at 2 M. When the initial concentration of oxalate in the dilute chamber was 0.4 M, 0.6 M, and 0.8 M, the corresponding concentration of oxalic acid in the concentrate chamber at the end of electrodialysis was C_OA_ (2 M HCl + 0.4 M OA, 50.02 mM) < C_OA_ (2 M HCl + 0.6 M OA, 76.13 mM) < C_OA_ (2 M HCl + 0.8 M OA, 82.34 mM), the corresponding concentration of hydrochloric acid in the concentrate chamber at the end of electrodialysis was C_HCl_ (2 M HCl + 0.4 M OA, 1219.02 mM) > C_HCl_ (2 M HCl + 0.6 M OA, 1030.23 mM) > C_HCl_ (2 M HCl + 0.8 M OA, 1019.33 mM), respectively. The results indicated that, when the HCl concentration was low in the mixed solution of the two acids, more oxalic acid could migrate across the membrane with the increase in oxalic acid concentration, thus reducing their separation effect by electrodialysis.

When the initial concentration of hydrochloric acid in the chamber was fixed as 3 M and 4 M, the effect of different initial oxalic acid concentrations (i.e., 0.4, 0.6, and 0.8 M) on the separation effect of two acids was also explored. Similar results were also found in Figure 4 when the initial concentrations of oxalic acid increased, which meant that more oxalate ions in the acid mixture participated in the competitive transmembrane migration with Cl^−^ during electrodialysis. When the concentration of hydrochloric acid in the mixed solution was higher, a higher concentration of hydrochloric acid was found in the concentrate chamber. The results in Figure 4 indicated that the higher molar ratio of hydrochloric acid to oxalic acid led to a higher proportion of chloride ions passing through the anion exchange membrane. According to the Appendix A, SAEM is composed of four distinctive layers (g). The fiber structure improves its mechanical strength, and particles are mixed in the layers (h). The membrane surface is smooth (k). CEM is a homogeneous membrane filled with particles (b). The surface has slight folds, and the whole is smooth and flat (e), while micropores appeared on the surface of the CEM after separation experiments (j).

#### 3.2.2. Ratio of Transport Number and Selective Transmittance Coefficient

The ratio of the transport number to the selective transmission coefficient of oxalate and Cl^−^ for the mixtures with different concentrations and molar ratios of hydrochloric acid and oxalic acid were further investigated. According to Figure 5a, when the concentration of hydrochloric acid in the dilute chamber was fixed at 2 M and the initial concentration of oxalic acid was 0.4 M, 0.6 M, and 0.8 M, respectively, the ratio of the transport number (t_Cl_^−^/t_OA_^−^) was approximately 20, 12, and 11, respectively, and the selective transmission coefficient of OA was approximately 0.2, 0.4, and 0.6, respectively. The results indicated that when the concentration of hydrochloric acid was mixed at 2 M HCl, the ratio of the transport number (t_Cl_^−^/t_OA_^−^) decreased gradually and the selective transmission coefficient of OA increased gradually with the increase in the OA concentration in the mixture, which was not conducive to the separation of oxalic acid and hydrochloric acid.

When the concentration of hydrochloric acid was fixed at 3 M or 4 M, respectively, it was found that for the higher initial concentration of hydrochloric acid, the ratio of the transport number (t_Cl_^−^/t_OA_^−^) decreased and the selective transmission coefficient of OA increased with an increasing concentration of oxalic acid from 0.4 M to 0.8 M in the mixture. The results showed that with the increase in the molar ratio of hydrochloric acid to oxalic acid, the ratio of the transport number (t_Cl_^−^/t_OA_^−^) increased and the selective transmission coefficient of OA decreased, indicating that the separation of hydrochloric acid and oxalic acid was better. The main reason was that the number of oxalate ions involved in the competitive migration with chloride ions decreases along with the increase in the molar ratio of hydrochloric acid and oxalic acid.

### 3.3. Separation Effect and Membrane Fouling of SAEM from Real Wastewater

The separation of hydrochloric acid from real rare earth oxalic acid precipitation mother liquor was also examined by electrodialysis with SAEM. The selective anion exchange membrane is ACS, and the cation exchange membrane is CSE. The mother liquor used in this separation was from the actual wastewater produced by a factory in Ganzhou, China. The chemical composition and water quality analysis of real rare earth oxalic acid precipitation mother liquor is shown in Appendix A. Combined with the rare earth industrial production and chemical composition analysis of the effluent shown in Appendix A, it could be inferred that the main impurities, such as the extraction agent (P507 [31]), heavy metal ions, hydrolysate (2–ethylhexanol, phosphoric acid [32]), diluent (sulfonated kerosene [33]), etc., might be contained in real wastewater. Thus, membrane fouling was also investigated when the real rare earth oxalic acid precipitation mother liquor was treated by electrodialysis with the SAEM.

Figure 6a showed that the concentration of hydrochloric acid in the concentrate chamber gradually increased and the concentration of hydrochloric acid in the dilute chamber gradually decreased during the whole ED process. Particularly in the early stage of electrodialysis at approximately 30 min, the concentration of hydrochloric acid in the dilute chamber decreased by more than 50%, while the concentration of oxalic acid in the concentrate chamber was still at a relatively low level (<15.46 mM) (shown in Figure 6b). The concentration of hydrochloric acid (~88.43 mM) in the dilute chamber at the end of ED was only 8% of the initial value (~1113.85 mM). The results revealed that hydrochloric acid could be effectively separated by electrodialysis with the SAEM. However, the migration rate of oxalic acid through AEMs was gradually faster along with the concentration decrease in hydrochloric acid contained in the real wastewater during the electrodialysis process. Moreover, the conductivity was evaluated in Appendix A. The conductivity of the dilute chamber decreased, and that of the concentrate chamber increased. These could be attributed to the migration of ions from the dilute chamber to the concentrate chamber, and the migration of chloride ions was the majority. The concentration of oxalic acid in the concentrate chamber increased suddenly after 35 min of electrodialysis, which meant that more oxalate ions in the acid mixture participated in the competitive transmembrane migration with Cl^−^. The limiting current density (*I_lim_*) of the SAEM was evaluated by separating the real rare earth oxalic precipitation mother liquid. The SAEM stack followed ohmic, non-ohmic, and over-limiting regions. Initially, the current density increased linearly with the applied potential of 4–12 V and followed Ohm’s law. At an applied potential of 12–14 V, the current density increased due to the rapid migration of ions through the membranes and depletion in the boundary layer. Thus, a large potential drop occurred, and the resistance of the system increased (plateau region). An over–limiting region is observed at the applied potential of >14 V due to water splitting [34]. The limit current density values of the SAEM stack ranged from 52.18 to 59.09 mA/cm^2^.

Combined with the rare earth industrial production and chemical composition analysis of the effluent shown in Appendix A, it could be inferred that the main impurities, such as the extraction agent (P507 [31]), heavy metal ions, hydrolysate (2–ethylhexanol, phosphoric acid [32]), diluent (sulfonated kerosene [33]), etc., might be contained in real wastewater. Thus, ion exchange membrane fouling was also investigated when the real rare earth oxalic acid precipitation mother liquor was treated by electrodialysis with the SAEM.

Compared with the original membranes of CSE and ACS, shown in Figure 7a,c, respectively, Figure 7b,d indicated that membrane fouling occurred after electrodialysis of the real rare earth oxalic acid precipitation mother liquor, in which the attachment layers of the organic matter and salt crystals were found on the surface of ACS and CES, respectively. The hydrochloric acid migrated into the concentrate chamber during electrodialysis, which led to a decrease in acidity in the solution of the dilute chamber. The dissolving equilibrium of rare earth oxalate in the dilute chamber might be broken when the acidity of the solution decreased. Thus, the rare-earth ions were likely to be combined with oxalate ions to form rare earth oxalate precipitates on the CSE surface. The organic compounds, such as P507, Phosphoric acid, 2–ethylhexanol, sulfonated kerosene, etc., contained in the real rare earth oxalic acid precipitation mother liquor could be complexed with rare earth or other heavy metal ions. Therefore, the organic compounds aggregated on the ACS surface via electrostatic interaction resulted in membrane fouling due to the formation of the organic adhesion layer. Both the ACS and CSE were rougher after use. The higher surface roughness of fouled membranes might be caused by inorganic salt deposition and the organic attachment layer, which was consistent with the SEM analysis.

The EDS analysis results of the fouled CSE surface shown in Figure 8a indicated that, compared with the original membrane, the atomic contents of Fe, Yb, Al, and Dy increased to 0.13%, 0.18%, 0.14%, and 0.03%, respectively, after membrane fouling. The cross–section EDS analysis shown in Appendix A indicated that the metal elements of Fe, Yb, Al, and Dy exhibited an even distribution in the cross-section of the fouled CSE, which meant that the CSE membrane could be fouled by rare earth and other heavy metals. The EDS analyses on the ACS membrane shown in Figure 8b indicated that the contents of P, S, Cl, and O elements increased to different degrees on the fouled ACS surface. The results in Figure 8b showed that the S element content of the fouled ACS surface increased from 2.73% to 6.02% that of the original ACS membrane. It was presumed that the organic matter containing sulfur and phosphorus in the solution, such as sulfonated kerosene and phosphoric acid, adhered to the surface of the ACS, resulting in an increase in the content of the corresponding elements. Moreover, all the elements of P, S, Cl, and O also exhibited an even distribution on the fouled ACS surface.

Different from the EDS analysis results of the CSE surface, the S content at the cross–section of the CSE membrane seems to be decreased, which indicated that the proportion of S in the cross–section was caused by other contaminating components entering the membrane inter–layer. The contamination experiment lasted several hours, and the pollution layer was very thin. Meanwhile, the ZnSe crystal beam splitter had a greater penetration depth capacity and could completely penetrate the fouling layer. EDS analysis was verified further by the fact that the peak strength of infrared –S=O did not change greatly. The elements of P, S, Cl, and O also exhibited an even distribution in the cross–section of the fouled ACS.

According to Figure 9a, it was found that the single-bond CH_2_ and single–bond CH_3_ did not chang much after fouling, which indicated the organic fouling on the surface of the CSE is relatively light. At 1140 cm^−1^ and 1040 cm^−1^, the characteristic peaks of –P-O and –S=O of the fouled ACS became much stronger, which was consistent with the result of the EDS analysis (shown in Figure 8b), and also indicated the contents of S, O, and P and other elements increased after ACS fouling. The ATR–FTIR spectra shown in Figure 9b indicated that the peaks at 2930 and 2860 cm^−1^ appeared on the spectra of the fouled ACS, which showed the single–bond CH_2_ and single-bond CH_3_ became much stronger. This could be closely related to the organic matter (such as P507, 2–ethylhexanol, etc.) contained in the rare earth oxalic acid precipitation mother liquor. This result demonstrated that the organic matter from the real rare earth oxalic acid precipitation mother liquor was adsorbed on the surface of the ACS during ED and caused serious ACS membrane fouling.

## 4. Conclusions

In this study, electrodialysis was used to separate the simulated hydrochloric acid and oxalic acid mixture. The performance of membrane materials, hydrochloric acid, and oxalic acid concentrations on the separation effect was investigated. The separation effect of SAEM was improved with the increase in the hydrochloric acid concentration in the mixture of two acids. The higher molar ratio of hydrochloric acid to oxalic acid was favorable for improving the separation performance. Similar results were also obtained in the real rare earth oxalic acid precipitation of the actual mother liquor treated by electrodialysis with the SAEM. Moreover, operating conditions of 15 V led to a high hydrochloric acid production purity (∼96.0%). Results indicated that hydrochloric acid could be effectively separated by electrodialysis with the SAEM, which provides a new method of resource utilization of oxalic acid and hydrochloric acid in the rare earth oxalic acid precipitation mother liquid.

Membrane fouling was investigated when the real rare earth oxalic acid precipitation mother liquor was treated. ATR-FTIR and SEM-EDS analysis showed that organic compounds containing carbon and sulfur were attached to the surface of the ACS, and salt crystals were also found on the surface of the CSE. Inorganic foulants of metal ions such as Fe, Yb, Al, and Dy were found in the CSE membrane section. Such pretreatment needs a systematic study, which is outside the scope of the present work.

## Figures and Tables

**Figure 1 membranes-13-00162-f001:**
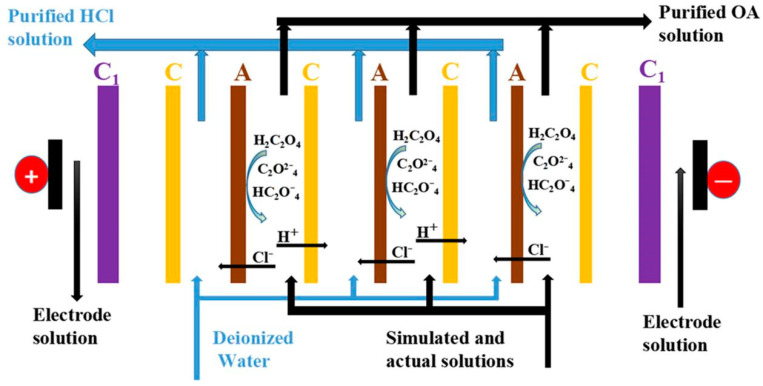
ED setup. C = cation exchange membrane, including three types of CAM, CTG−10, and CSE; A = anion exchange membrane including three types of ATG−10, AAM, and ACS; C_1_ = Electrode membrane.

**Figure 2 membranes-13-00162-f002:**
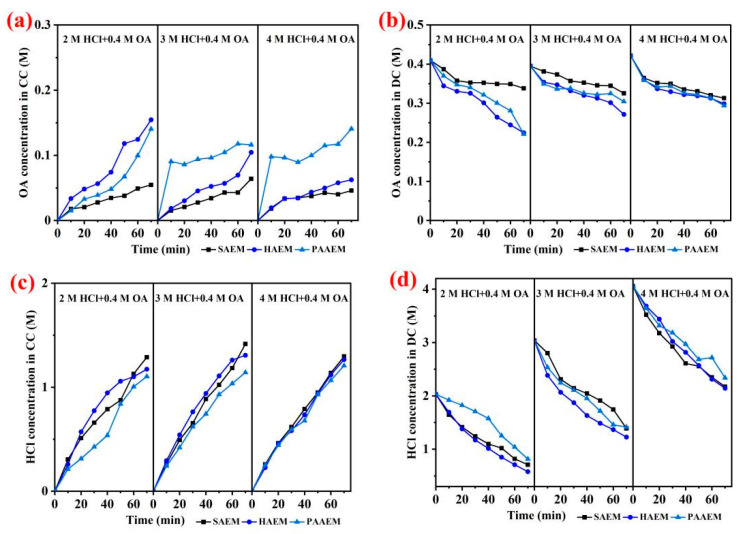
Oxalic acid concentration curves in concentrate chamber (i.e., OA concentration in CC) and dilute chamber (i.e., OA concentration in DC) with the AEMs were shown in (**a**,**b**); hydrochloric acid concentration curves in concentrate chamber (i.e., HCl concentration in CC) and dilute chamber (i.e., HCl concentration in DC) were shown in (**c**,**d**).

**Figure 3 membranes-13-00162-f003:**
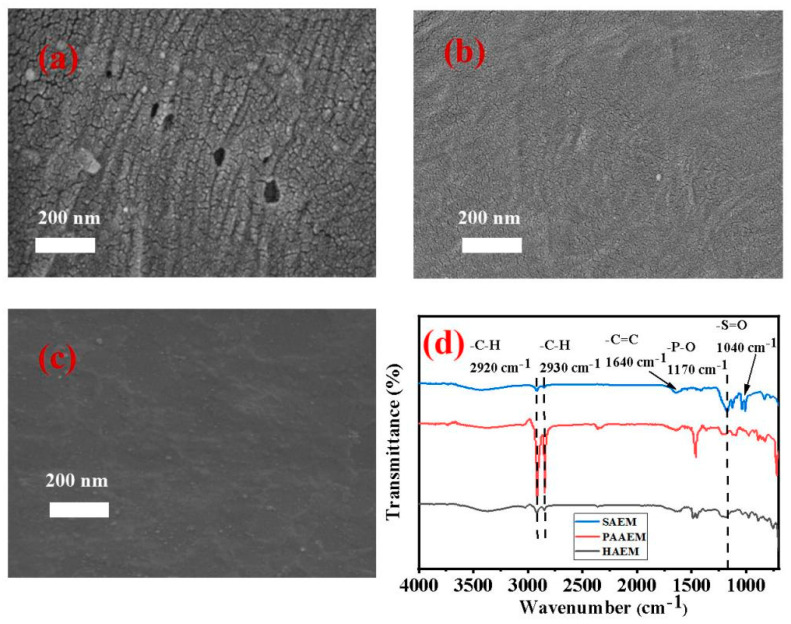
SEM images of PAAEM (**a**), HAEM (**b**), and SAEM (**c**); (**d**) ART-FTIR analyses of three AEMs.

**Figure 4 membranes-13-00162-f004:**
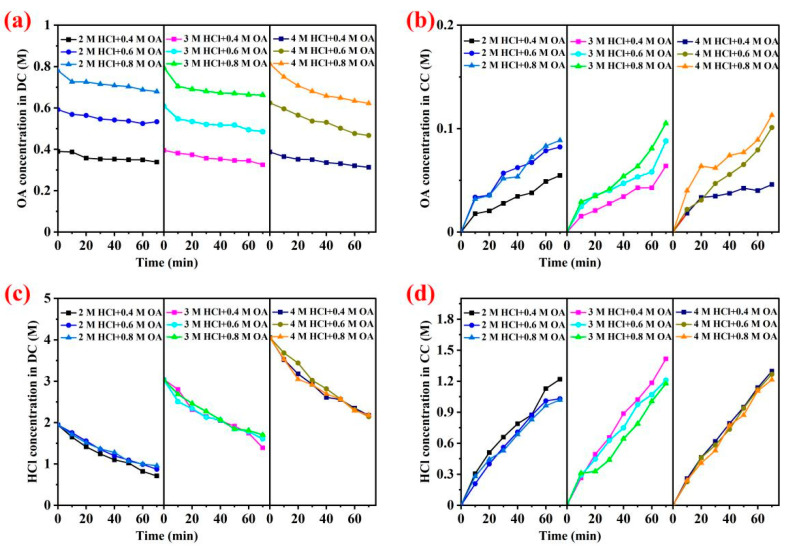
Oxalic acid concentration curves in dilute chamber (i.e., OA concentration in DC) and concentrate chamber (i.e., OA concentration in CC) are shown in (**a**,**b**); hydrochloric acid concentration curves in dilute chamber (i.e., HCl concentration in DC) and concentrate chamber (i.e., HCl concentration in CC) are shown in (**c**,**d**).

**Figure 5 membranes-13-00162-f005:**
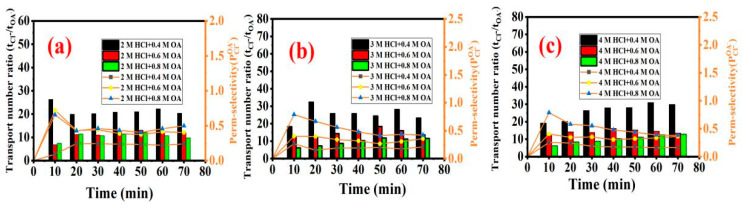
(**a**–**c**) The ratio of transport number (t_Cl_^−^/t_OA_^−^) and OA selective transmission coefficient in the different acid mixtures of 2 M HCl +0.4/0.6/0.8 M OA, 3 M HCl +0.4/0.6/0.8 M, and 4 M HCl +0.4/0.6/0.8 M OA, respectively.

**Figure 6 membranes-13-00162-f006:**
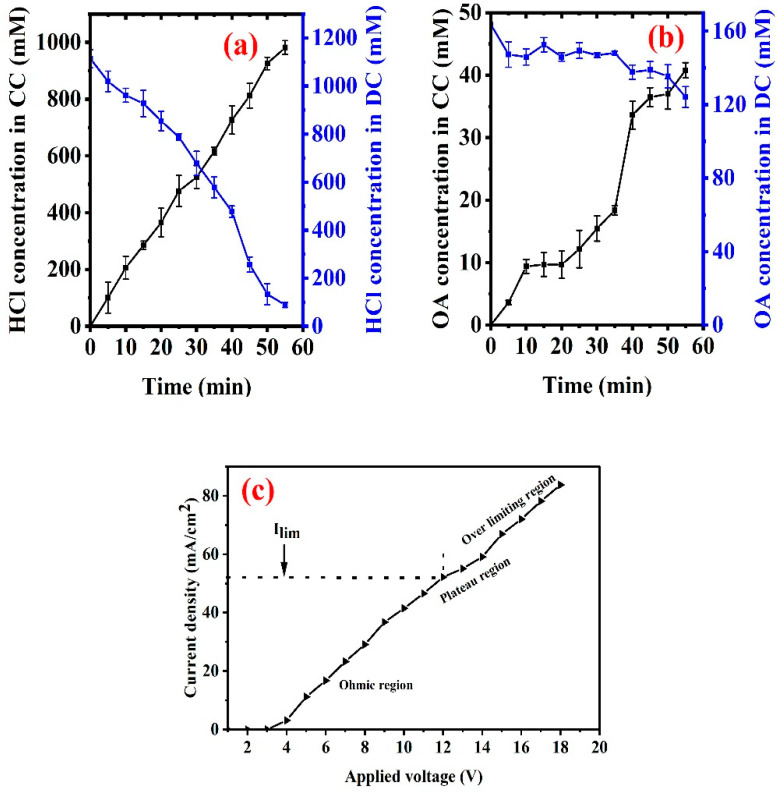
(**a**) The concentrations of hydrochloric acid in concentrate chamber (i.e., HCl concentration in CC) and dilute chamber (i.e., HCl concentration in DC); (**b**) the concentrations of oxalic acid in concentrate chamber (i.e., OA concentration in CC) and dilute chamber (i.e., OA concentration in DC). (**c**) The current density–voltage curves.

**Figure 7 membranes-13-00162-f007:**
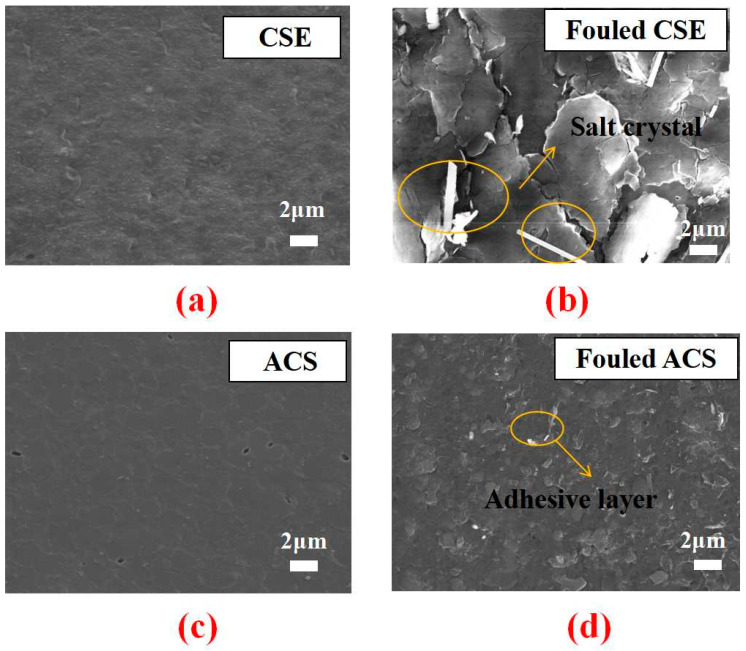
SEM images of CSE and ACS fouled before (**a**,**c**) and after (**b**,**d**).

**Figure 8 membranes-13-00162-f008:**
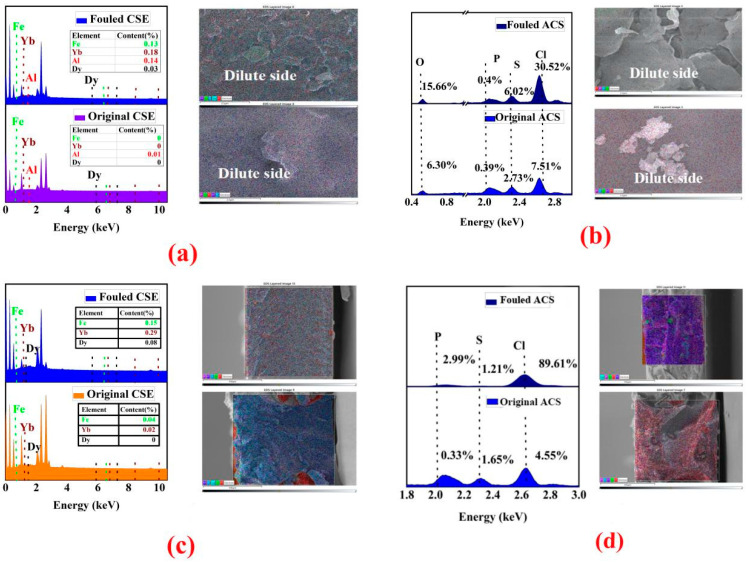
SEM–EDS images and Fe, Yb, Al, Dy, O, P, S, and Cl elements mapping content of membrane surface of CSE and ACS are shown in (**a**,**b**); SEM–EDS images and Fe, Yb, Dy, P, S, and Cl elements mapping content of membrane cross–section of CSE and ACS are shown in (**c**,**d**).

**Figure 9 membranes-13-00162-f009:**
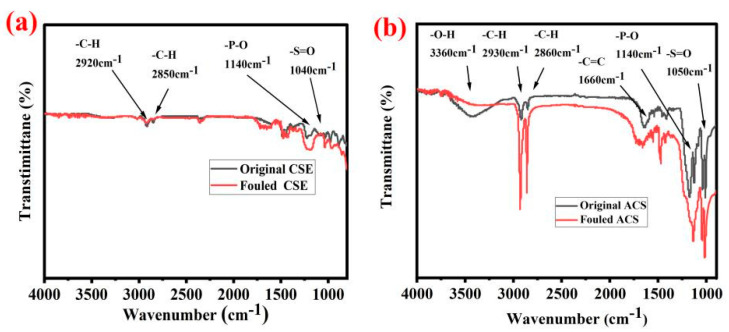
ATR–FTIR analyses of ACS (**a**) and CSE (**b**) (used before and after).

**Table 1 membranes-13-00162-t001:** The main performance parameters of the ion exchange membrane used in this study.

Parameter	Unit	Ion Exchange Membranes					
		ACS	CSE	AAM	CAM	ATG-10	CTG-10
Thickness ^1^	mm	0.13	0.16	0.36	0.32	0.18	0.19
Resistance ^2^	Ω cm^2^	3.8	1.8	6	7	2.3	1.9
Bursting strength ^3^	MPa	≥0.15	≥0.35	0.6	0.6	1	0.5
Using temperature ^4^	℃	≤40	≤40	≤40	≤40	≤40	≤40
Effective Area ^5^	m^2^	0.055	0.055	0.055	0.055	0.055	0.055
pH ^6^	--	0~8	0~14	2~12	2~12	2~10	2~10
Water absorption ratio ^7^	%	20.59%	36.14%	26.62%	50.94%	23.48%	36.36%

Parameters 1–6 were obtained from the official website of membrane manufacturers. All of the membranes used in this study contain C, H, N, O, and Na elements, and styrene diethylbenzene. Resistance was measured using alternating current at 25 °C in 0.5 M NaCl solution. Parameter 7 was calculated (shown in Appendix A).

## Data Availability

Data is contained within the article or Appendix A.

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
