# Peer review of "Separation of Hydrochloric Acid and Oxalic Acid from Rare Earth Oxalic Acid Precipitation Mother Liquor by Electrodialysis"

_membranes, 2023, doi:10.3390/membranes13020162_

Round 1

Reviewer 1 Report

1.       “In the study the hydrochloric acid from rare earth oxalic acid precipitation mother liquor 12 was separated by electrodialysis (ED) with different anion exchange membranes, including selective 13 anion exchange membrane (SAEM), polymer alloy anion exchange membrane (PAAEM) and ho-14 mogenous anion exchange membrane (HAEM).” Check grammar please

2.       “The results indicated that the hydrochloric acid could be separated effectively by elec-17 trodialysis with SAEM from simulated and real rare earth oxalic acid precipitation mother liquor, 18 in which the hydrochloric acid obtained in the concentrate chamber of ED is of higher purity gen-19 erally.” Mention numbers for the purity

3.       Define ACS, CSE

4.       “Now oxalic acid is a 39 commonly used precipitant in the purification of rare earth (RE) from a concentrated preg-40 nant leach solution (PLS) [4], in which the H+-ions contained in oxalic acid are released 41 and combined with Cl- in concentrated solution of RE to form hydrochloric acid with high 42 concentration.” Check grammar please

5.       “The effluents can only be discharged after proper treatment, otherwise the high COD and high acids contained in the waste liquid will cause serious 46 environmental pollution.” Discharged to where?

6.       “The performance of membrane dis-55 tillation is limited by low temperature and concentration polarization leading to vapor 56 pressure reduction [11].” Concentration polarization also occurs in ED and may be more serious.

7.       “Moreover, long time operation of membrane distillation system 57 will lead to membrane scaling, pore clogging and rate reduction due to the impurities 58 contained in wastewater [12].” These drawbacks are also present in ED!!

8.       “Another unique advantage of electrodialysis is the 65 selective separation the univalent ion of Cl−) versus multivalent ion of SO42‐ with monova-66 lent permselective AEMs due to their selective permeability for the two ions [16-22].” Check grammar and are you separating CL- from SO4?

9.       “The separation performance of three different types of AEMs including SAEM, HAEM, 71 PAAEM for oxalic acid and hydrochloric acid was compared.” Why do you use different types of membranes?

10.   Add references to all mentioned mathematical equations.

11.   What is the operating voltage and ampere?

12.   Figure 2 organization is complex. Please simplify and improve.

13.   In Figure 2, mention the direction of curves as for in DC or CC.

14.   “The results indicated that, when the HCl concen-210 tration was low in the mixed solution of the two acids, more oxalic acid could migrate 211 across the membrane with the increase of oxalic acid concentration, thus reducing their 212 separation effect by electrodialysis.” What effect?

15.   Remove dot from “4. Conclusion”

16.   The conclusion is not meaningful. The present conclusion seems like a summary of the results.

Reviewer 2 Report

The manuscript requires major revisions before it might be considered for publication.

 -Introduction. Membrane processes should be mentioned as promising sustainable processes to concentrate REEs (see e.g. Sep. Purif. Technol. 254 (2021) 117641).

-Section 2. The chemical composition of the various membranes should be provided.

-The term “intercepted” (several places in the manuscript; see for example line 167) is not appropriate.

-Fig. 3a. Holes of about 20 nm can be seen of the PAAEM membrane surface. How the membrane can still be selective to ions?

-Lines 193-194. I am a bit surprised that -P-O and -S=O functional groups, usually associated with the presence of phosphonic and sulfonic acid groups, were detected on the surface of the SAEM membrane as the latter is an anion exchange membrane.

-Units for concentration should be homogenized throughout the manuscript (the authors mixed M and mmol ; see e.g. lines 205-209).

- The origin and composition of the treated solutions should be clearly indicated in section 2. It is only from line 300 onwards that the reader learns about the presence of kerosene, sulphur, phosphorus...

-Lines 311-315. What was the thickness of the fouling layer compared to the penetration depth of the IR beam?

Reviewer 3 Report

1. The introduction needs more discussion about previous studies in the field or REE precipitation and electrodialysis, and clear statement of novelty of research

2. Line88- protection form what? (for sodium bisulfite)

3. The thickness of membranes sounds very low 130 nm

4. Please present results as transport number instead of migration number. Transport number is commonly accepted language

5. How is SAEM performance best in Figure 3? Trend shown in graph is different from the absolute values presented.

6. line 166- What do you mean oxalic acid "intercepted"? Do you mean ions? Cl- versus oxalate ion? Please be clear

7. Line 181- how does pore structure cause low oxalic acid interception?

Round 2

Reviewer 2 Report

The author have addressed my comments properly. The revised manuscript can be published.

Reviewer 3 Report

Comments addressed